# Impact of COVID-19 on mental health of primary healthcare workers in Pakistan: lessons from a qualitative inquiry

Waqas Hameed [1], Bilal Iqbal Avan [2], Anam Shahil Feroz,[1] Bushra Khan,[3] Zafar Fatmi [1], Hussain Jafri,[4] Mansoor Ali Wassan,[5] Sameen Siddiqi [1]

[1]Community Health Sciences, The Aga Khan University, Karachi, Sindh, Pakistan
[2]Department of Population Health, London School of Hygiene and Tropical Medicine, London, UK
[3]Department of Psychology, University of Karachi, Karachi, Sindh, Pakistan
[4]Punjab Thalassaemia and other Genetic Disorders Prevention and Research Institute, Fatima Jinnah Medical University, Lahore, Pakistan
[5]Department of Health, Government of Sindh, Karachi, Pakistan

**Correspondence to**
Dr Bilal Iqbal Avan;
Bilal.Avan@lshtm.ac.uk

## ABSTRACT

**Objectives** The existing literature regarding the mental health consequences of COVID-19 among healthcare workers revolves predominantly around specialised hospital settings, while neglecting primary healthcare workers (PHCW) who are the first point of contact for patients. In view of negligible evidence, this study explored the mental health impact of COVID-19 and health system response, and sought suggestions and recommendations from the PHCWs to address their mental health needs during the pandemic crisis.

**Design** We employed a qualitative exploratory design.

**Setting** A total of 42 primary healthcare facilities across 15 districts in Sindh and Punjab provinces of Pakistan.

**Participants** We telephonically conducted 47 in-depth interviews with health service providers and hospital managers. A combination of inductive and deductive approach was used for data analyses using NVivo V.11.0.

**Results** There was immense fear, stress and anxiety among PHCWs being infected and infecting their families at the beginning of this outbreak and its peak which tapered off over time. It was triggered by lack of information about the virus and its management, false rumours, media hype, lack of personal protective measures (personal protective equipment, PPE) and non-cooperation from patients and community people. Trainings on awareness raising and the PPEs provided by the healthcare system, with emotional support from coworkers and supervisors, were instrumental in addressing their mental health needs. Additionally, they recommended appreciation and recognition, and provision of psychosocial support from mental health professionals.

**Conclusion** Primary healthcare system should be prepared to provide timely informational (eg, continuous updates in training and guidelines), instrumental (eg, provision of PPE, appreciation and recognition), organisational (eg, safe and conducive working environment) and emotional and psychosocial support (eg, frequent or needs-based session from mental health professionals) to PHCWs in order to mitigate the mental health impact of pandemic crisis.

## STRENGTHS AND LIMITATIONS OF THIS STUDY

⇒ Our study gathered the mental health experiences of a neglected group of healthcare workers that belonged to primary healthcare settings.
⇒ Unlike urban-focused studies, the participants in this study were invited across 15 urban and rural districts in two most populous provinces of Pakistan.
⇒ Conducting phone-based interviews due to the restricted mobility during the COVID-19 was a key study limitation which may introduce some risk of bias as the interviewers did not have the opportunity to build rapport as in the case of face-to-face interviews.

## INTRODUCTION

COVID-19 has swept across the world.[1] As of 28 September 2021, 232 million people have been infected by COVID-19 and approximately 4.75 million people have lost their lives worldwide,[2] including 115 000 healthcare workers (HCWs).[3] This large-scale unprecedented public health catastrophe placed the front-line HCWs, who are directly involved with all aspects of care of patients with COVID-19, at risk of developing psychological distress and other mental health symptoms.[4] Studies conducted thus far provide evidence of the adverse impact of COVID-19 on mental health and well-being of HCWs, primarily those who are involved in providing specialised care to patients with COVID-19.[5] According to a systematic review, the pooled prevalence of depression, anxiety and post-traumatic stress disorder among HCWs related to the COVID-19 pandemic across 21 (low to high income) countries hovered around 22%.[5] The prevalence of such mental health disorders varies according to sex, age and role of HCWs.[6 7]

While these studies made substantial contribution to inform strategies in alleviating the burden of psychological distress among HCWs, the primary focus has been on professionals who are working in hospital settings and are primarily engaged in the provision of specialised care to patients with COVID-19.[6 7] In contrast, primary healthcare workers (PHCWs) have been managing a major share of the burden of COVID-19-related care[8] by

being the first point of contact for a broad spectrum of patients with COVID-19, whether suspected, confirmed (with or without symptoms) or exposed to case(s), as well as those with non-COVID-19 health needs.[9] Studies indicate that some 80% of cases are mild and the majority of moderate cases seek primary healthcare (PHC) services as the entry point for getting medical care.[10]

Community health workers (CHWs) also played a distinct role during the COVID-19 in lower middle-income countries by being involved in community awareness, engagement and sensitisation, and contact tracing though at the cost of routine service delivery.[11] These newly assigned tasks during the pandemic exposed them to increased risk of infection, and they experienced stigmatisation, isolation or social ostracisation.[11] Ironically, they reportedly received inadequate support from health system in the form of personal protective equipment (PPE)[12] and logistics and supportive supervision. Also, PHCWs are also responsible in managing the aftermath of contingencies and bearing the collateral damage caused by reallocation of resources to COVID-19 while inadvertently hindering access to a wide range of PHC services.[13] [14] This put them under stressful situations during this pandemic.[15] A study from Canada showed that interpersonal care team rapidly shifted their practices to support their patients during the COVID-19 pandemic, and they experienced a surge in mental health issues among their patients.[16] Similarly, a large-scale survey from Canada revealed that COVID-19 resulted in increased workload, loss of employment, redeployment to new setting and concerns for health and safety among social workers at the front line.[17]

Despite the ongoing debate on how strengthening the PHC (including Community Health Workers, CHWs) can help health systems adapt during the COVID-19 pandemic,[18] [19] the significance and instrumental role of PHCWs during the COVID-19 pandemic have received little attention from the global research community. A large majority of the research that assessed the mental health consequences of COVID-19 among HCWs has focused on the specialised hospital settings.[20] Furthermore, there are few qualitative studies that provide in-depth mental health experiences of HCWs in general, as well as those working in PHC settings.[21] Even a recent effort to consolidate potential interventions to address mental health issues of HCWs during such outbreak is largely based on evidence generated from HCWs working in specialised settings, while neglecting the mental health needs of and solutions for PHCWs.[22]

Keeping in view of the role of PHCWs as alluded to above, we hypothesised that they may have suffered from similar mental health issues faced by HCWs working in specialised healthcare settings. Furthermore, as COVID-19 is expected to become endemic,[23] [24] the role of PHC becomes pivotal for its community-based information structure to identify risk groups, monitor compliance and provide preventive and promotional services as well as contact tracing in this and similar future

pandemic. Hence, their mental health needs, readiness and response become central to tackling such issues. To better support PHCWs with their adapted role during the COVID-19, interventions are needed that could effectively cater to their mental health (which include mental, emotional and social) needs in the long run.[25] With this rationale, we set the following questions for this research that focuses on public-sector PHCWs in Pakistan.

## Research questions

1. What is the perceived mental health impact of COVID-19 on PHCWs in Pakistan?
2. How does health system respond to the mental health needs of PHCWs?
3. What are the recommendations to support the mental health and well-being of PHCWs during the pandemic crisis?

## Country context

Pakistan has a population of over 225 million inhabitants.[26] The first COVID-19 case was identified in February 2020, and since then a total number of confirmed cases that exceed 1.2 million along with 27 638 deaths have been registered (until 28 September 2021).[27] The country has experienced four waves—each with different variant including Delta, more recently.[27] With a focus on vulnerable groups of HCWs and elderly (65+ years) people, the country embarked on a nationwide vaccine campaign for COVID-19. The vaccine is now available for everyone except for children under the age of 12 years.[27] While several vaccines are safe and effective, there is strong emphasis on practising preventive measures by the government.

Public-sector healthcare in Pakistan is delivered through a three-tiered health system. The PHC provides services through a community-based component comprising over 100 000 CHWs and health facility component that includes almost 10 000 basic health units (BHUs), rural health centres (RHCs), maternal and child centres (maternal and child health) and civil dispensaries.[28] On average, BHU and RHC cover a population of 25 000 and 80 000, respectively. These static PHC facilities are supplemented by approximately 100 000 CHWs (referred to as 'lady health workers' (LHW)).[29] The government of Pakistan has taken several measures to combat this deadly pneumonia virus,[30] but it leaves a lot to be desired.[31] The health system of Pakistan is weak,[32] has critical shortage of health workforce and is not prepared to sustain a major surge in COVID-19 cases.[33] Few studies have been conducted to assess the impact of COVID-19 on mental health of HCWs; while these studies identify symptoms of mental disorders, all have focused on the tertiary level of health system.[34–36]

## METHODS

### Study design and setting

A qualitative exploratory design was adopted using an in-depth interview (IDI) technique. Exploratory design enabled us to gather a detailed description of HCW experiences during the COVID-19 pandemic. The study was conducted in 42 health facilities across 15 districts of Sindh and Punjab provinces of Pakistan. Districts where the cumulative number of COVID-19 cases was high since the onset of this outbreak until date of survey were selected. Participants were selected using a purposive sampling approach criterion from randomly selected public health facilities stratified by districts and type of health facilities (BHUs and RHCs). We also collected the contact details of affiliated LHWs[29] from those in charge of the selected health facility.

### Recruitment of study participants

The study participants were classified into three groups: LHWs, facility-based health service providers (doctor, nurse, midwife, lady health visitor) and administrators/managers (medical superintendent, lady health supervisors (LHS)). LHWs are attached to a local health facility, but they are primarily community based, working from their homes. They serve the whole community, but they play a particularly important role in maternal and child healthcare in rural and urban slum communities by coordinating efforts with traditional birth attendants and midwives.[37] These LHWs work under the supervision of LHS.[38 39] In contrast, the facility-based HCWs are based at PHC facilities and they provide the following services: general outpatient services, antenatal care, deliveries, postnatal care and vaccination. In addition to these, diagnostic services (laboratory, X-ray) and dental outpatient services are also provided in RHCs. The PHCWs were primarily involved in awareness raising and contact tracing regarding COVID-19. Some of the facility-based staff were also seconded to COVID-19 isolation centres. Even though administrators at the PHC were also often directly involved in service provision, we kept this distinction to see any diversity in their responses as opposed to their counterparts. The study participants were recruited after obtaining formal permission from the higher government authorities by contacting the administrators of the randomly selected health facilities.

### Data collection

A standard semistructured guide was used to interview all study participants, which was developed following extensive review of literature and consultation. It comprised three major themes: (a) impact of COVID-19 on mental health and well-being of healthcare professionals; (b) mechanism available within the healthcare system to address mental health needs; and (c) mental health needs of PHCWs, as well as suggestions and recommendations to address their needs. Open-ended questions and probes encouraged free flow of information from participants to develop a deeper understanding of their experiences.

The interview guide was thoroughly reviewed and discussed by the core research investigators (WH, BIA, ZF, ASF, BK and SS). It was then pretested and revised based on few PHCWs in districts that were included in the larger quantitative survey.[40–42] During pretesting, no major current challenges were reported by most PHCWs since the first wave has already passed.[42] Therefore, the guideline was revised to gather their experiences from the onset of COVID-19 to current situation, and getting insights on how situation evolved over time (see online supplemental file 1).

The data collection took place between August and October 2020. Owing to restricted mobility due to COVID-19, a total of 47 IDIs (Sindh: 29; Punjab: 18) were conducted telephonically until data saturation was achieved.[43] In the first phone call, the participant was informed about the study and was scheduled an appropriate date and time for the interview, if the participant is busy. Subsequent call was made at the scheduled time for the interview. Experienced data collectors with a background in sociology or psychology were trained to conduct the interviews in local language (Urdu). All the calls were made from The Aga Khan University under close supervision of study investigators. Data analysis was concurrently performed to determine the point of saturation. On average, the interviews lasted for 25–35 min. With the consent of participants, all the interviews were audio recorded on tablets using a built-in call recording feature. It was ensured that participants were comfortable during the interview and no one was sitting near them whose presence could possibly influence their responses. Data collectors were also encouraged to maintain key interview notes. A linguistic expert later transcribed the recordings into English for analysis. The transcripts were not returned to the participants due to their busy schedule. Five participants refused to participate in the study due to their busy schedule; they were mainly the health managers. No repeat interview was conducted. To ensure data quality, a few initial interviews were conducted by WH (PhD scholar—health system) and ASF (PhD scholar—public health) in the presence of trained data collectors. Both authors were university faculty at the time of the study and had vast experience in conducting such interviews.

### Data analysis

All interview notes, recordings and transcriptions were imported into the NVivo V.11.0 software for analyses, with no identifying characteristics included in the transcriptions.[44] We explored the divergent and congruent views of the three groups, that is, facility-based service providers, administrators and LHWs. Conventional content analysis technique was used to inductively analyse all data.[45] Two independent researchers (WH and ASF) applied inductive coding while reviewing the transcriptions in NVivo[44] and identified new emerging codes and categories. The codes and themes were compared and any discrepancies in the interpretation of codes and themes were discussed

among the investigation team to reach a consensus. The discrepancies were discussed in a large team meeting until an agreement on the themes was achieved. In situations when it was difficult to achieve consensus among the team, the senior author facilitated the discussions and helped obtain agreements among the research team members. To gain a more comprehensive understanding of mental health impact, health system response and suggestions and recommendation to support their mental health needs, subthemes were reviewed and compared by additional reviewers (BK and ZF) to examine the corroboration and convergence. Independent review of data and coding allowed coherent interpretations that are grounded in the data.[41] We did not obtain feedback from the participants on study findings.

## Patient and public involvement

We sought inputs from target audience through pretesting of interview guides to ensure comprehension of the questions. Furthermore, in consultation with provincial health departments we selected districts where COVID-19 cases were high. The focal person in each selected district was oriented about the study and connected our team with those in charge of health facility to avoid any miscommunication. Lastly, they also advised about the dissemination of research findings—that is, they specifically emphasised to share it through research briefs to reach a wider audience in the country.

## Ethical considerations

Due to the telephone interview, verbal informed consent was taken from all study participants which included permission to audio record their interviews and use anonymised quotes. All study participants were informed about the study purpose, their rights as study participants, confidentiality and publication of study findings in anonymised manner. We shared with study participants the official permission letters issued by the respective health department of the government of Punjab and Sindh provinces to build trust.

The Consolidated Criteria for Reporting Qualitative Research guidelines were used to organise this paper[46] (see online supplemental file 2).

## RESULTS

A total of 47 IDIs were included in the analysis. Table 1 shows the sociodemographic characteristics of participants. The mean age of the participants was 43.2 (±10.9) years and experience was 14.4 (±9.4) years. Approximately half were female (44.7%), 55.3% were health managers and about 51.1% were working at or affiliated with the BHU. Those in charge of health facilities (BHU and RHC) were all medical doctors and mostly males.

The main study findings are organised into three main themes: perceived mental health impact of COVID-19; support mechanisms to address their mental health needs; and suggestions and recommendations made by

**Table 1** Characteristics of study participants

| Characteristics | n (%) |
|---|---|
| Sex | |
| Female | 21 (44.7) |
| Male | 26 (55.3) |
| Age in years | |
| Mean (±SD) | 43.2 (10.9) |
| Designation | |
| Health service providers (doctor, nurse/midwife) | 16 (34.0) |
| Health managers (lady health supervisors—supervise LHWs, in charge of health facility) | 26 (55.3) |
| Lady health workers (LHWs) | 5 (10.6) |
| Type of health facility | |
| Basic health unit (BHU) | 24 (51.1) |
| Rural health centre (RHC) | 23 (48.9) |
| Total experience in years | |
| Mean (±SD) | 14.4 (9.4) |

PHCWs. Where observed, we also presented contrasting views of service providers and health managers, highlighting the diversity in provincial perspective (table 2).

## Mental health impact of COVID-19 on HCWs

### Anxiety due to uncertainty of COVID-19 and media exaggeration

All study participants reported experiences of immense fear and anxiety due to the uncertainties around COVID-19 at the onset of this outbreak in Pakistan. It was primarily triggered by lack of knowledge, prevalent misconceptions about COVID-19 and being aware of its disastrous effect in the developed world. There was a perception that it was a fatal virus and whoever got it would die. Others were sceptical about its existence, like how long this outbreak will last and if they will remain bound forever to the new operational changes in service delivery.

> At the beginning, the real fear among people was that—a rumour spread that, if God Forbids, this much dangerous disease, and someone gets it he will die. (Health manager, Ghotki)

> There's a bit of worriedness, mental pressure, what to do what not to do, how to go, should we go to their house or not, these problems happens, mental stress occurs. (LHW, Jamshoro)

A large majority of HCWs were not able to distinguish between COVID-19-positive and negative people around them due to their asymptomatic characteristics and lack of testing facility which was mentally disturbing. It made the HCWs reluctant to interact with patients and community members, and even with their colleagues at the very beginning of the outbreak. This was consistent among facility-based service providers, health managers and LHWs.

**Table 2** Themes and categories

| Themes | Categories |
|---|---|
| Theme 1: perceived mental health impact of COVID-19 on healthcare workers | 1.1 Anxiety due to uncertainty of COVID-19 and media exaggeration.<br>1.2 Fear of infection, isolation and stigma.<br>1.3 Challenges regarding patients.<br>1.4 Stress due to lack of availability of personal protective equipment.<br>1.5 Stress due to excessive workload. |
| Theme 2: available support mechanisms to address mental health needs of healthcare workers | 2.1 Awareness raising of healthcare workers.<br>2.2 Provision of personal protective equipment.<br>2.3 Social support from coworkers and family members.<br>2.4 Managerial support from supervisors. |
| Theme 3: mental health needs of healthcare workers and recommended solutions | 3.1 Safe and conducive working environment.<br>3.2 Awareness about evolving situation and corrective actions.<br>3.3 Psychosocial support from mental health professionals.<br>3.4 Appreciation and recognition of their work during the pandemic crisis. |

We ourselves were also mentally disturbed, that how do we know, we are going on duties, on fields, we staffs are meeting each other so how do we know that who has corona symptoms and who does not. (LHS, Rawalpindi)

### Fear of infection, isolation and stigma

While most of the PHCWs were not directly involved in providing care to patients with COVID-19, there was still some fear of being contracted with COVID-19 during their interaction with patients at the beginning and during peak time. Similar fear existed among LHWs during their routine household and targeted visits for contact tracing.

We were afraid that we might get infected or our families might get infected. So we spend all this time under stress. We were seeing that there were many positive cases in our neighbourhood and the news on media. (Health manager, Shaheed Benazirabad)

Some of the facility-based staff were seconded to COVID-19 isolation centres. It was highlighted that the level of fear was considerably higher among them. Similarly, the level of fear of possible transmission was higher among HCWs who had any chronic disease (eg, diabetes) or had old parents or young children, be they working at the health facility or in the community.

At that time, my female health workers were very disturbed mentally. She was like we have small kids how we will run out-patient services… how everything will be managed. Their families were also tensed that you are going on duties and you are putting us on danger as well. (Health manager, Shaheed Benazirabad)

These stressors and uncertainties of being infected also affected the provider–patient interaction during the initial phase of the outbreak and as it peaked. A service provider shared that:

When a person is mentally unwell and is feeling uncertain, then s/he wouldn't even deliver what is normally expected of him/her. We were not able to take care of the patients then. We didn't know whether the incoming patient is COVID positive or not. We just kept a distance and asked patient ailment. We had to save ourselves someway. (Service provider, Faisalabad)

Fear of isolation was not reported by many HCWs. Only a few facility-based service providers reported to be stressed as they had to stay away from their family members, or by being alone.

She [my co-worker] took her medicine regularly for 14 days at home. Then her [COVID-19] test was conducted after 14 days and it came negative. But still she was very worried and tensed, and she was crying that she had to stay away from her kids. (Service provider, Khairpur)

Mixed responses were reported by the LHWs in terms of stigma faced by them. According to some LHWs, community members resisted their entry into the households, whereas senior or experienced LHWs reported fewer such issues in the field.

During corona [peak] days, when we [I and my LHW] used to go out, people used to complain a lot. A huge pandemic has spread and you are coming to our house, so how do we know whether you have it [COVID-19] or not; you can infect us. We felt this kind of resistance a lot in field even though we used to go with PPEs (sanitizers and masks)… at some places, sometimes it [people's reaction] was normal and some people gave very positive responses. (LHS, Rawalpindi)

### Challenges regarding patients

A major challenge that was a source of stress and frustration among HCWs was uncooperative patients. Facility-based staff were concerned that patients were reluctant to wear face mask or to adhere to physical distancing within the premises of health facilities, and at times they

became offensive when they were insisted to take protective measures for COVID-19.

> Our stressor was our patients because they were not ready to understand the situation. They didn't cooperate with us. They thought we didn't want to treat them. We get the announcement done here in our local areas and also pasted the posters. We also wrote on the board outside to 'wash your hands' and 'wear mask in hospital'; but we had to tackle a lot of troubles like these. (Health manager, Gujrat)

Due to illiteracy, people did not believe in the existence of any viral outbreak, and they considered such protective measures as tactics of health workers to avoid service delivery.

> Only those women are worried who are a little bit educated but mostly aren't, 99% of people are don't have any fear and they don't believe in corona existence and they think that we are making them stupid. (Health manager, Mirpur Khas)

Early on, another challenge faced by facility-based staff (both service providers and managers) was to conduct COVID-19 test for suspected patients. According to them, people were extremely reluctant to share any details or willing to undergo COVID-19 test. This was primarily due to the prevalent myths and misconceptions regarding COVID-19 at the beginning of this outbreak.

> People used to bring sticks to beat us because it is a rural area and villagers used to say that our patients didn't have anything and they asked us to go away. We had to call assistant commissioner of that area than he used to send police. After that we were able to take samples from them. It was mental torture for us. Believe me, when we tell people that we are about to take their samples, tell us your mobile number and Identification Card number etc. I swear they curse us and run away. (Service provider, Sargodha)

### Stress due to lack of availability of PPEs

Lack of availability of PPEs or supplies put both facility-based and community-based HCWs under immense stress of being infected. Even facility-based HCWs, who were seconded to a quarantine centre, expressed concerns about unavailability of PPEs.

> I had duty in quarantine for 3 months then I didn't have any kit or PPE, or mask or glove. I got a call and they told me to come on duty that day at 2 so I started crying because both my parents were home. We heard about doctors and XYZ deaths. I was scared and didn't have any protection kit. (Service provider, Sialkot)

In the absence of PPEs, some of the facility-based service providers and LHWs made use of the scarf or linen to cover themselves and their patients' face or a community member during household visit, respectively,

while majority of them reported to have it purchased on their own.

> Obviously, there weren't many masks available for us even and for patients. So any piece of cloth, or scarf, we use to make them cover their mouths and nose. Males sometimes used linen or face mask. (Health manager, Ghotki)

### Stress due to excessive workload

Increased workload was mainly the concern of field-based staff (LHWs and LHS) and those who were given additional responsibilities like secondment to COVID-19 isolation centre. Field teams had to locate suspected cases at a short notice. Lack of proper transportation, especially during lockdown, made things quite stressful and exhaustive.

> Everybody was busy running to and fro because [COVID-19] cases would come without notice. Doctor would get anxious and ask us to collect data and pictures [of patients]. We had to go to LHS and LHW. (LHS, Rawalpindi)

On the contrary, the workload of facility-based service providers reduced considerably during the initial phase in the health facilities as people were reluctant to go to health facilities in fear of COVID-19 and certain prevalent myths.

> Yes, it (health facility) was working but patients were very less. Usually we have 200 or 300 patients but during corona we had only 30 or 40 patients. Even there was news circulating in news channels that poison injection were being given in the hospitals to corona patients. So people were really afraid and due to these kinds of rumours they stopped coming to hospitals. (Service provider, Sargodha)

### Available support mechanism to address mental health needs of health workforce

#### Awareness raising of HCWs

All facility-based and community-based HCWs reported to have received trainings on awareness raising that were organised by the department. The trainings primarily focused on preventive aspects of COVID-19: characteristics of virus, signs and symptoms, modes of transmission, population at risk, preventive and protective measures, quarantine guidelines and its testing.

> The purpose of training was to screen yourself, keep yourself safe, save the people of your organization, and also protect community; things that can control the disease. If a patient has symptoms, tell them where to do test and how it is done. (Service provider, Khairpur)

However, participants from all three study groups unanimously informed that they had not received any training on mental health except for one LHS.

In trainings, we talked on all issues, right. But, the main issue that was not touched is people's mental condition during corona. Nobody ever tried to assess that amongst us. We have never ever tried to understand people mental level or their mental condition or our feelings. (LHS, Rawalpindi)

According to all HCWs, increased awareness and declining COVID-19 cases over time mitigated the fear and stress among HCWs.

Right now [health] workers are not depressed they are normal and doing their routine work. They were depressed and worried in the beginning but now they are normal due to awareness campaigns. (Health manager, Shaheed Benazirabad)

### Provision of PPEs

PPEs were provided to the health workers with some delays, after few weeks of the outbreak in the country. The use of these PPEs, combined with other protective measures of COVID-19, played a major role in reducing fear and stress among HCWs.

We displayed the posters outside of hospitals—informing people not to come to the hospital without any reason. Maintain social distancing, wash hands, make soaps available everywhere and made hand sanitizers available. That's how we coped up with the stress. To be very frank, in the beginning everything was very difficult for me and for them [health workers]. (Service provider, Multan)

Firstly, protection, masks, shields etc. there was a shortage before but now they are available. There is also awareness now, we are seeing that there are fewer patients in Pakistan compared to other countries. All these things are effective for mental health. (Service provider, Sialkot)

### Social support from coworkers and family members

Provision of social support by coworkers—including colleagues and supervisors—emerged as a common response mechanism to mitigate the fear, stress and anxiety among both facility-based and community-based HCWs. During the early phase of this outbreak when stress, anxiety and fear were high, staff members used to discuss and provide social support to each other. Senior HCWs often used to enquire about challenges being faced by their colleagues, including if they were emotionally disturbed. They used to console them to help overcome any kind of fear, stress or anxiety pertaining to COVID-19.

We just provided them emotional support. I educated staff members that we are health care providers, how could we help people if all of us become a victim of mental health issues. We have to be strong because we are on frontline. We don't have to be fearful. Along with that we used to appreciate workers. What if they themselves become depressed then who would work.

I told them that corona has no limit so we will have to live with it. (Health manager, Khairpur)

A couple of HCWs acknowledged that they used to consult and take frequent advice from ex-physiatrist mates to overcome stress, and some received support from their family members.

I was in contact with a psychiatrist so in that way I used to get constant help from there. I personally took many sessions. (Service provider, Faisalabad)

My husband kept on telling me that the night that is in the grave is not outside, so stop thinking about it and the major support that my husband gave me was it is a death by martyrdom so I used to get relaxed from that as well. (LHS, Rawalpindi)

The emotional support was provided in several ways. Some of the consoling approaches were: (a) reinforcement of the use of PPEs and COVID-19 standard operating procedures (SOPs); (b) reiterating the purpose of being a health worker; (c) ensuring frequent communication with the stressed person and giving assurance of being with them; (d) motivating colleagues by giving their own example; and (e) encouraging others to keep themselves busy if they are stressed. Importantly, some used religious healing perspective by linking everything with God 'Allah', that He will take care of everything.

All you have to do is care, and this is a type of help and if any patient gets well with our help it would be a blessing from Allah, living and dying is Allah's decision, we can only take care so take care and help others, you will be rewarded InshaAllah [if Allah wills] so we say all this to each other and we give support to each other. (Service provider, Dadu)

I was continuously in contact with her [COVID-19 positive health worker] and tell her to take care of herself and she was telling her children whether test will come negative or not. She was scared that she had corona and she would die. Yes, we told her God will help you. We have doctors with us and our district manager and medical superintendent are with us, they were talking to her continuously. They were also consoling here that corona will end why you are worried? (Service provider, Khairpur)

Reassuring, giving trust, and with positive thinking for the future, telling them that we are messengers, Allah has placed us to serve these people, to take care of them, to tell them preventions. (LHS, Rawalpindi)

A few facility-based HCWs kept themselves busy in extra-curricular activities to deal with their stress, like reading or even social media.

I kept myself engaged because whenever I was free then I would get a lot stress when you hear news etc. I was studying and spent most of my time with my co-worker. And whenever I would go home, I would keep myself engaged mentally. (Service provider, Sialkot)

## Managerial support from supervisors

According to a majority of service providers and LHWs, their supervisors/managers provided a lot of support in managing work-related issues that were causing stress. The supervisors primarily managed issues of their teams by themselves, or at times, by engaging higher (district) administrative health authorities. Effective communication with team that included constant motivation and awareness raising was instrumental in addressing day-to-day challenges. At some instances, health managers also offered intermittent breaks to the health workers who were unwell or became exhausted due to work. A couple of respondents also mentioned that the health facility was closed for a few days when the COVID-19-positive staff were quarantined.

> We fulfil their demands, if there is a salary issue, if they haven't received their salary from 2 to 3 months, we write it to DHO that they will not do work with interest so they should be given their salary on time and they forward it to upper department and their problems are solved. (Service provider, Dadu)

> I counsel them that see this is difficult for us; you cooperate with me and I will cooperate with you. If you have any problem then you can tell me and take off. If there is no issue then take half day off but come to work next day. So I tried to make them relax like this. (Health manager, Karachi)

## Mental health needs of HCWs and recommended solutions
### Safe and conducive working environment

Safe working environment was identified as one of the prominent needs of both facility-based and community-based HCWs. It primarily meant timely provision of PPE kits that should include gloves, face masks and hand sanitisers, and provision of security especially in dealing with aggressive patients.

> We should have received PPE kits in the beginning before [COVID-19] peak time. When cases were increasing in China, since then we should have made ourselves prepared for this. (Service provider, Multan)

> Security should be given especially in this COVID situation. Because when patients come to the health facility and the doctor fails to examine them then they create a scene or become aggressive, and misbehave. (Service provider, Sialkot)

Primarily, facility-based HCWs who were assigned additional responsibilities related to COVID-19 expressed a dire need of intermittent breaks under certain circumstances—for example, when a worker is ill, exhausted due to increased workload or experiences any kind of stress.

> If someone appears COVID-19 positive then that person is placed in 14 days quarantine. So if he is already in a stressful condition, his mental health isn't able to hold so at least until he is normalized, there should be some relief given. (Service provider, Faisalabad)

> They [health workers] say that how can we perform our normal duties and COVID ward both simultaneously? So they should be given holidays. Our demand is that our staff should be given more holidays and good packages and some financial bonuses should be provided. (Health manager, Mirpur Khas)

### Awareness about evolving situation of COVID-19 and corrective actions

A majority of facility-based and community-based HCWs expressed the need for having updated information about the evolving situation regarding COVID-19. They also identified the need of receiving timely guidelines about the corrective actions in dealing with constantly changing situation.

> There should have been sessions about the forthcoming epidemic. Like this what is this disease, what are its consequences, how to deal with it. There should've been such learnings that they prepare their workers. They just said that 'wear masks', and what's being told on TV. (Health manager, Khairpur)

### Psychosocial support from specialised mental HCWs

Even though social support emerged as a key response measure used by HCWs to cope with the prevailing stress, getting psychosocial help from specialised mental health professional was still identified as a need and future recommendation by majority of the study participants. Various suggestions were given to have such support in place for the HCWs, which included: (a) having a full-time specialist-based district or subdistrict level to provide needs-based support, (b) conducting periodic sessions on mental health, (c) mobile team of psychologists to provide periodic support for health staff and (d) making mental health component a part of the COVID-19 trainings.

> Mental health is a drawback here. There should be a proper person, who should, first of all, educate the staff, that how will you improve the mental health of the people. Then, some training should be arranged there, meaning, there should be a proper person sitting there, right, like some psychiatrist, or some people like these. (LHS, Rawalpindi)

> In my opinion, CEO of our hospital should depute someone from DHQ like 3 sessions in a week so that they can give sessions to healthcare workers, perform counselling and try to know their inner problems due to which they are disturbed so that their problems can be solved. And provide them relief if it is related to salary or any other issue. (Service provider, Sargodha)

Some facility-based HCWs suggest use of mobile application to provide information about mental health and well-being.

They should conduct seminars and this is the era of multimedia everyone has mobile and there should be more content on multimedia and social media about mental health so that we become mentally strong and cater things properly. There should be seminars and sessions and government should launch an app regarding this so that we get information from it regarding mental and physical health and keep on updating it. (Health manager, Gujrat)

On the other hand, facility-based HCWs expressed the need of capacity building to provide adequate psychosocial support to coworkers and to their patients, as needed. These trainings are also expected to help them guide the general patients or community people to follow COVID-19 SOPs.

No, patients aren't believing, that's why psychiatrist is important and there should be training provided by the psychiatrist. Our staff should be provided psychological training, we should know what they [patients] are saying and what their opinions are so then we can handle them accordingly. (Health manager, Mirpur Khas)

### Appreciation and recognition of their work during the pandemic crisis

Both facility-based and community-based HCWs demanded recognition and appreciation for the additional work that they had to do during the COVID-19 crisis. This need was particularly highlighted by those who were given additional responsibilities like secondment at isolation centres, contact tracing and COVID-19 testing of suspected cases. This recognition was demanded in the form of monetary incentive for risking their lives, or a letter of appreciation by the health department, or a continuous encouragement from their administrative leadership.

Lady Health Workers are already doing extra work, putting their lives in danger, should have gotten some extra benefits for this, even if they had gotten any certificate. Even if we get one praise, they want to sacrifice everything. (LHS, Rawalpindi)

I would say that government should facilitate the doctor, increase the allowance, give them a complete personal kit, if someone dies, government should support their family, so this should be on government level, so doctors also work motivated… I would want that government should give full support, financially and morally. (Service provider, Khairpur)

### DISCUSSION

The current body of existing literature revolved predominantly around the mental health and well-being of HCWs who are involved in the provision of specialised COVID-19-related health services. It largely neglects the PHCWs who are the first point of contact for patients and are also involved in a wide range of COVID-19-related activities, which in turn can impact their mental health. Our study is unique in that it explored the mental health impact of COVID-19 and health system response, and sought suggestions and recommendations from the PHCWs to address the mental health needs of PHCWs during the pandemic crisis.

Reportedly, constant fear, anxiety and stress of being infected and infecting others were immense at the onset of COVID-19 outbreak and during its peak time. This was primarily triggered by the lack of information about the virus and its management, false rumours regarding COVID-19, media hype, lack of availability of PPEs and not able to distinguish between patients who are positive and negative for COVID-19. Evidence shows that exaggeration by media tends to negatively affect the mental health of HCWs.[47 48] Similarly, lack of knowledge about COVID-19 infection is likely to increase anxiety/depression among physicians,[49] and inadequate precautionary measures in the workplace cause fear and anxiety among HCWs.[50]

Another important concern causing distress and frustration among HCWs was non-compliance with COVID-19 SOPs by incoming patients and their companions. According to service providers, low literacy in rural populations, lack of awareness and non-serious attitude towards COVID-19 were the primary reasons for this non-adherence. This is consistent with another study from Pakistan that identified non-compliance with COVID-19 SOPs as one of the reasons for stress and anxiety among HCWs.[35] Misconceptions and stigma attached to COVID-19 also made it immensely challenging for HCWs for contact tracing and COVID-19 testing on suspected patients. In extreme cases, people became aggressive and reacted violently to HCWs. Similar incidents have been reported in Pakistan and other countries.[51 52] Such incidents have been associated with work-related stress, burnout and post-traumatic stress disorder, which in turn negatively affect job performance and quality of services.[53]

The mental health impact of COVID-19 tapered off over time among PHCWs. PHCWs attributed the reduction of stress and anxiety to the support provided by the government in the form of awareness raising sessions regarding COVID-19 and its preventive measures, as well as provision of PPE. More importantly, social support by coworkers (including supervisors) emerged as a key factor that alleviated the level of stress and fear among HCWs at the beginning when it was very high. Social support by coworkers has also been reported as an effective response mechanism in dealing with prevalent mental health issues among HCWs during the COVID-19 crisis.[54 55] Given its significance, fostering a systematic mechanism of social support within the healthcare system has also been recommended to safeguard the mental health of HCWs during the pandemic.[55] Consistent with other studies,[9] patient visits by PHCWs were significantly reduced due to fear of COVID-19 infections among the general population. Workload was only a concern of PHCWs who

were assigned additional task regarding COVID-19, and these HCWs reported experiences of exhaustion due to additional responsibilities. They also demanded recognition and appreciation in the form of official appreciation letter or monetary incentives for risking their lives. This finding is consistent with other studies where HCWs expressed similar needs.[18 51]

In the end, PHCWs rendered few recommendations to address their mental health needs during the pandemic crisis. First, they expressed the necessity of having a safe and conducive working environment that included timely provision of PPE for all staff members, safety and security of HCWs in dealing with difficult patients and intermittent breaks—especially for those who were assigned additional COVID-19-related responsibilities. The positive effect of safe and conducive working environment on mental health is already discussed above. Second, recommendation was around preparedness of HCWs through continuous awareness raising and corrective actions about the evolving pandemic situation. Increased knowledge is also linked with reduced level of stress and anxiety among HCWs.[49] Third, they highlighted the need for psychological counselling from mental health professionals and suggested a different approach to fulfil this need. Call centres have been established in many countries—including Pakistan[56]—to provide needs-based psychosocial support to HCWs. In Pakistan, this programme perhaps needs to expand their scope and provide more targeted services to the PHCWs. Additionally, HCWs requested the need for training to be able to provide psychosocial support to coworkers and in dealing with difficult patients or people in the communities. There has already been a debate about the potential role of CHWs in the provision of psychosocial support to people during the pandemic crisis.[57] The existing 'WeCare' programme of the Pakistani government could also be effectively integrated with PHC settings to assist HCWs in providing support to the incoming patients. Lastly, a key need identified by the HCWs, especially those who were assigned additional tasks related to COVID-19, was appreciation and recognition for their work during the pandemic crisis. They particularly stressed on developing a formal mechanism for the acknowledgement of their efforts via appreciation letter or in the form of monetary incentive or risk allowances. A similar need was also identified in other studies as well.[58]

## Strengths and limitations

In terms of strengths, our study used gathered experiences of a neglected group of HCWs that belonged to PHC settings. It uses an inductive interpretative approach to achieve the study objectives by gathering information from diverse group of service providers and health managers, including LHWs. Importantly, this research revealed the perspective and experience of HCWs from the rural settings which are largely neglected in the current body of knowledge. It allowed the researchers to look for converging and diverging lines of inquiry to identify common themes/concepts and incongruence between data sources. On the contrary, a major limitation was that all interviews were conducted over the phone which may have constrained rapport building with participants and obtained non-verbal cues during the interviews. In addition, the interviewees might have been distracted during interviews by the activities in their environment and might have experienced fatigue due to long telephone interviews. Another limitation is that the study was conducted with a focus on public-sector hospitals in only two provinces of Pakistan (Sindh and Punjab), which would limit the transferability of the findings to other provinces.[59] However, this study may provide insights for similar PHC settings across Pakistan that are interested in understanding the mental health impact and mental health needs of health service providers during the COVID-19. Lastly, the study was unable to carry out member checking with study participants, which may have affected the validity of study findings.

## CONCLUSION

Our study concludes that the stress and anxiety among PHCWs were immense at the beginning of this outbreak or when the COVID-19 transmission was at its peak. However, it tapered off as they become more aware of the virus and receive PPEs, and as the number of COVID-19 cases decreased in their community. The mental health of either health workers or patients was not covered in the trainings offered to them—neither had they access to any professional psychosocial support. Our study identified a need for a comprehensive support that should be provided to PHCWs in a timely manner, including informational support (eg, continuous awareness about evolving situation and corrective actions); safe and conducive working environment (eg, provision of PPE, personal security in dealing with violent reactions of community people due to stigma); organisational support (eg, intermittent breaks, appreciation and recognition in the form of appreciation letters or monetary incentives (especially for those who were assigned additional responsibilities)); and frequent professional psychosocial support to cope with stress but also capacity building trainings on mental health that will enable them to deal with difficult patients or community people.

**Acknowledgements** We are grateful to our implementing partners (Department of Health, Government of Sindh; Primary and Secondary Healthcare Department (P&SHD); Specialized Healthcare and Medical Education Department from Government of Punjab) for extending their full support and guidance to conduct this survey. We are also thankful to all our study participants for sparing their precious time and making themselves available for the interviews. Last but not the least, we thank our data collectors and project management team (Zafar Dehraj, Imran Sheikh, Sajid Brohi, Noreen Afzal, Ghani Muhammad) who managed the implementation of this study in a proficient manner during the pandemic.

**Contributors** WH and BIA conceptualised the study with inputs from BK, ZF and SS. WH, BIA, BK, ZF and ASF developed the interview guide. HJ and MAW facilitated the conduct of the study. WH and ASF performed the data analysis. BK and ZF reviewed again the themes and subthemes. WH took the lead in writing the manuscript. All authors critically reviewed the manuscript and provided feedback in

shaping the write-up, analysis and presentation of results. WH is the overall content guarantor.

**Funding** This work was cofunded by The Aga Khan University, Pakistan (grant number: 20051) and WHO (grant number: 202568710-1).

**Competing interests** None declared.

**Patient and public involvement** Patients and/or the public were involved in the design, or conduct, or reporting, or dissemination plans of this research. Refer to the Methods section for further details.

**Patient consent for publication** Obtained.

**Ethics approval** This study involves human participants and was approved by The Aga Khan University (ID: 2020-5186-11602) and the National Bioethics Committee of Pakistan (reference number: 4-87/COVID-45/NBC/20/393). Participants gave informed consent to participate in the study before taking part.

**Provenance and peer review** Not commissioned; externally peer reviewed.

**Data availability statement** No data are available.

**ORCID iDs**
Waqas Hameed http://orcid.org/0000-0002-8100-9474
Bilal Iqbal Avan http://orcid.org/0000-0003-4531-4508
Zafar Fatmi http://orcid.org/0000-0001-7212-6858
Sameen Siddiqi http://orcid.org/0000-0001-8289-0964

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
