## [Reviewer comments · BMJ Open]

ARTICLE DETAILS

TITLE (PROVISIONAL)	Impact of COVID19 on mental health of primary healthcare workers in Pakistan: lessons from a qualitative inquiry
AUTHORS	Hameed, Waqas; Avan, Bilal Iqbal; Feroz, Anam Shahil; Khan, Bushra; Fatmi, Zafar; Jafri, Hussain; Wassan, Mansoor Ali; Siddiqi, Sameen

VERSION 1 – REVIEW

REVIEWER	Indiana-Luz Rojas-Torres Universidad Simón Bolívar
REVIEW RETURNED	03-Aug-2022

GENERAL COMMENTS	1. Recruitment of study participants: Please expand the information on the characteristics or roles of the study subjects, especially: lady health workers and health service providers in establishments.2. Interview guide: Please expand this information for the proposal of this structured guide of questions, review process, etc. results obtained in the pilot test3. Please, expand information on the process for consensus in the interpretation of codes4. Please expand on the information analysis taking into account not all the participants were performing the same functions and it is understood that the needs are different.5. Could you provide information based on the three problem questions to each study group? questions problems/group study subjects
--

REVIEWER	Rachelle Ashcroft University of Toronto
REVIEW RETURNED	21-Aug-2022

GENERAL COMMENTS	This study explored the psychological impact of COVID19, health system response, and sought suggestions and recommendations from the primary healthcare workers (PHCWs) to address their psychological needs in pandemic crisis. Hence, this study is asking what are PHCWs psychological needs resulting from the COVID-19 pandemic? The manuscript interchangeably refers to psychological needs, mental health, and psychosocial throughout yet is a bit confusing given that these are different concepts. The first two RQ's focus on psychological impact, yet the study justification paragraph centres on mental health. It's not to say that each of these three concepts and impacts are not important – they definitely are for this PHCW population – however, it is not clear how they are being used distinctly in the background and/or as foundations for the study.
--

	The first two research questions ask about psychological impact, and supports related to psychological needs. Then, the third RQ asks about recommendations to support mental health and psychosocial needs. Shouldn't the third RQ ask about recommendations to support/improve psychological state and/or wellbeing? The first two questions weren't focused on mental health nor psychosocial needs. Greater continuity across the RQs would be beneficial. Background: -Good justification -States that the virus has affected 232 million people. Is this intended to mean that 232 million people have been infected by covid-19? If not, then clarify what "affected" means in this regard. Affected could mean many things. -Clarify if the stat on page 4, line 6 is stating that 115,000 healthcare workers have died d/t COVID-19. Line 13, page 4: "According to a systematic review, the pooled prevalence of depression, anxiety and post-traumatic stress disorder among HCWs across 21 (low to high income) countries hovered around 22%". Make it explicit that this is related to the COVID-19 pandemic. How does this compare to pre-pandemic? Background: More information is needed on the role of PHC in the pandemic. Page 4 describes PHC being essential and experiencing burnout, but does not talk about the role of PHC during the pandemic (although talks about the role of community health workers below this paragraph without having described role of PHC). The following manuscripts presented early data on the role of PHC, and the impact of the covid-19 pandemic on primary care teams, including burnout, which would help bolster the background related to PHC & covid-19 & the discussion: -Donnelly et al., 2021; Ashcroft et al., 2021. -What is the definition of PHC guiding this study? Add definition. Also, there is both primary care, and PHC used in the manuscript. Additionally, "mental health" and "MH" are both used throughout. -What is the definition of Community health worker in this study? -Country context is described well. Context of Study: -42 health facilities across 15 districts in Sindh and Punjab provinces of Pakistan Methods: -Qualitative descriptive design. -Individual interviews for data collection, conducted via telephone -The section titled "Interview guide" should be a sub-heading under "Data Collection" -The concepts in the interview guide don't align with the research questions. -Who was the interview guided pilot tested on? Explain what comparable non-study means.
--	---

	-Page 4, line 47: It is unclear if the participant sample referred to as “Lady Health Workers” are those healthcare workers who are providing women’s health care services, or, are these female participants? If the participants are female-identified participants, then it is not clear why participants of one gender would be identified and targeted distinctly. -Recruitment of study participants: This section needs more clarity. It is not clear who the study participants are. More details are needed on how participants were recruited. -Patient and public involvement: States that authors’ sought inputs from target audience through pre-testing of interview guides. Be more specific in explain this – more details are needed. -Engagement of provincial health departments is an asset & strength of the study. Results: What is “IDIs”? Results: -It is unclear how the breakdown of characteristics of the participants relate to PHC. The designation of type of provider is unclear as is the type of health facility. More details are needed. Data analysis: Indicates that content analysis is used for data analysis, although the presentation of findings don’t necessarily appear to reflect content analysis.
--	---

VERSION 1 – AUTHOR RESPONSE

Reviewer 1 Comments to the Authors:

Dr. Indiana-Luz Rojas-Torres, Universidad Simón Bolívar
Comments to the Author:

Comment 2.1: Recruitment of study participants: Please expand the information on the characteristics or roles of the study subjects, especially: lady health workers and health service providers in establishments.

Response 2.1: Thank you. As suggested more details have been added. Please see page 5 and 6.). Just to elaborate it here that: Lady health workers are attached to a local health facility, but they are primarily community-based, working from their homes. They serve the whole community, but they play a particularly important role in maternal and child health care in rural and urban slum communities by coordinating efforts with traditional birth attendants and midwives. On the other hand, the facility-based healthcare workers based in primary healthcare facilities usually provide following services: general OPD, antenatal care, deliveries, postnatal care, and vaccination. In addition of these, diagnostic services (lab, x-ray) and dental OPD are also provided in rural health centres. The PHCWs were primarily involved in awareness raising and contact training regarding COVID19. Some of the facility-based staff were seconded to COVID19 isolation centres.

Comment 2.2: Interview guide: Please expand this information for the proposal of this structured guide of questions, review process, etc. results obtained in the pilot test

Response 2.2: We have elaborated the section on interview guide development, pilot testing and its revision.

Comment 2.3: Please, expand information on the process for consensus in the interpretation of codes

Response 2.3: We have elaborated the process for consensus in the manuscript. Just to summarise here that “The main themes and sub-themes were identified independently by the primary researcher and second reviewer and then discussed in the large team meeting until agreement on the themes was achieved. In situations when it was difficult to achieve consensus among the team, senior author facilitated discussions and helped obtain agreements among the research team members”

Comment 2.4: Please expand on the information analysis taking into account not all the participants were performing the same functions and it is understood that the needs are different.

Response 2.4: Thank you for your comment. We have treated participants as separate groups based on their responsibilities. Essentially, we had two groups: service providers (nurse, midwife, doctor, community health workers) and managers (in-charge of health facility and lady health supervisors). Taking into account participants' responsibilities, we have carried out our analysis and presented results accordingly – that is, highlighting any divergent views of participant in particular group. This has been further highlighted in the revised results section at various instances.

Comment 2.5: Could you provide information based on the three problem questions to each study group? questions problems/group study subjects

Response 2.5: By and large, the experiences were similar across the three group. However, as suggested, we have elaborated more on the divergent view across the groups of study participants. Please see our edits on page 8, 9, 10, 11, 12, 13, 14, 15.

Reviewer 2 Comments to the Authors:

Dr. Rachelle Ashcroft, University of Toronto

Comments to the Author:

Comment 3.1: This study explored the psychological impact of COVID19, health system response, and sought suggestions and recommendations from the primary healthcare workers (PHCWs) to address their psychological needs in pandemic crisis. Hence, this study is asking what are PHCWs psychological needs resulting from the COVID-19 pandemic?

Response 3.1: Yes, this is the primary purpose of our study.

Comment 3.2: The manuscript interchangeably refers to psychological needs, mental health, and psychosocial throughout yet is a bit confusing given that these are different concepts. The first two RQ's focus on psychological impact, yet the study justification paragraph centres on mental health. It's not to say that each of these three concepts and impacts are not important – they definitely are for this PHCW population – however, it is not clear how they are being used distinctly in the background and/or as foundations for the study.

Response 3.2: Many thanks for highlighting an important point. Psychological needs, mental health, and psychosocial are distinct yet overlapping concepts. Primarily, we wanted to understand the health staff's psychological need that would encompass their mental, emotional and social needs. To avoid confusion, we are now using the terms 'psychological needs' throughout the manuscript. We used the term mental health as it is commonly used to indicate psychological issues.

Comment 3.3: The first two research questions ask about psychological impact, and supports related to psychological needs. Then, the third RQ asks about recommendations to support mental health and psychosocial needs. Shouldn't the third RQ ask about recommendations to support/improve psychological state and/or wellbeing? The first two questions weren't focused on mental health nor psychosocial needs. Greater continuity across the RQs would be beneficial.

Response 3.3: We have rephrased the research questions (RQ) a little to bring more clarity. The revisions are made in view of our response to comment 3.2 above.

Background:

Comment 3.4: Good justification

Response 1.4: Thank you for your appreciation.

Comment 3.5: States that the virus has affected 232 million people. Is this intended to mean that 232 million people have been infected by covid-19? If not, then clarify what “affected” means in this regard. Affected could mean many things.

Response 3.5: We have revised the sentence.

Comment 3.6: Clarify if the stat on page 4, line 6 is stating that 115,000 healthcare workers have died d/t COVID-19.

Response 3.6: Yes, we confirm. Please see the source URL:

<https://www.who.int/news/item/20-10-2021-health-and-care-worker-deaths-during-covid-19>

Comment 3.7: Line 13, page 4: “According to a systematic review, the pooled prevalence of depression, anxiety and post-traumatic stress disorder among HCWs across 21 (low to high income) countries hovered around 22%”. Make it explicit that this is related to the COVID-19 pandemic. How does this compare to pre-pandemic?

Response 3.7: Thank you. We have made it explicit that this is related to the COVID19 pandemic. Comparing these estimates with those from the WHO on common mental disorders among the global population, at 4.4% for depression and 3.6% for anxiety disorders (including PTSD), highlights the substantial impact of the COVID-19 pandemic on the psychological wellbeing of health care workers.

Reference: Depression and Other Common Mental Disorders: Global Health Estimates. Geneva: World Health Organization; 2017.

Comment 3.8: Background: More information is needed on the role of PHC in the pandemic. Page 4 describes PHC being essential and experiencing burnout, but does not talk about the role of PHC during the pandemic (although talks about the role of community health workers below this paragraph without having described role of PHC). The following manuscripts presented early data on the role of PHC, and the impact of the covid-19 pandemic on primary care teams, including burnout, which would help bolster the background related to PHC & covid-19 & the discussion:

-Donnelly et al., 2021; Ashcroft et al., 2021.

Response 3.8: Thank you very much for your suggestion. We have included relevant information in the background and discussion section of our manuscript from the suggested articles.

Comment 3.9: What is the definition of PHC guiding this study? Add definition. Also, there is both primary care, and PHC used in the manuscript. Additionally, “mental health” and “MH” are both used throughout.

Response 3.9: The description about PHC and roles of study participants have been added in the methods section. Please see page 5 & 6. We have consistently used the term PHC throughout the manuscript after writing it in full (i.e. primary healthcare). With respect to the use of mental health and MH, we have consistently used the term ‘psychological’ in place of mental health or psychosocial. This was pointed out by another reviewer since psychosocial, psychological, and mental health are conceptually different terms.

Comment 3.10: What is the definition of Community health worker in this study?

Response 3.10: We have provided detailed about community health workers in our study and also provided a brief description of her role in Pakistan. This was also suggested by another reviewer. Please see page 5 and 6.

Comment 3.11: Country context is described well.

Response 3.11: Thank you very much for your appreciation.

Context of Study:

Comment 3.12: 42 health facilities across 15 districts in Sindh and Punjab provinces of Pakistan

Response 3.12: Yes, this is correct.

Methods:

Comment 3.13:-Qualitative descriptive design.

Response 3.13: Yes, this is correct.

Comment 3.14:-Individual interviews for data collection, conducted via telephone

Response 3.14: Yes, this is correct.

Comment 3.15:-The section titled “Interview guide” should be a sub-heading under “Data Collection”

Response 3.15: The ‘interview guide’ has been moved under ‘Data Collection’.

Comment 3.16:-The concepts in the interview guide don’t align with the research questions.

Response 3.16: We have tweaked the description under interview guide in accordance with the research questions.

Comment 3.17:-Who was the interview guided pilot tested on? Explain what comparable non-study means.

Response 3.17: This qualitative study was conducted concurrently with the larger quantitative survey. This interview was pre-tested on few primary healthcare workers (similar target audience like lady health workers, nurses/midwife etc.) who belonged to the districts that were not included in this qualitative study but part of the quantitative survey. The details have been added in the manuscript. Please see page 7.

Comment 3.18:-Page 4, line 47: It is unclear if the participant sample referred to as “Lady Health Workers” are those healthcare workers who are providing women’s health care services, or, are these female participants? If the participants are female-identified participants, then it is not clear why participants of one gender would be identified and targeted distinctly.

Response 3.18: The ‘Lady Health Workers’ are essentially ‘community health workers’. They all are female workers. We have provided more details on their roles and responsibilities. Please see page 5 and 6.

Comment 3.19: -Recruitment of study participants: This section needs more clarity. It is not clear who the study participants are. More details are needed on how participants were recruited.

Response 3.19: We have elaborated that section. A brief description about participants has been added along with the recruitment strategy. Please see page 5 and 6.

Comment 3.20: -Patient and public involvement: States that authors’ sought inputs from target audience through pre-testing of interview guides. Be more specific in explain this – more details are needed.

Response 3.20: Thank you very much for your suggestion. We have appended COREQ checklist along with the manuscript as a separate file.

Comment 3.21: -Engagement of provincial health departments is an asset & strength of the study.

Response 3.21: Thank you for highlighting this. Yes, this was done in collaboration with health departments.

Comment 3.22: Results: What is “IDIs”?

Response 3.22: IDIs is an abbreviation of ‘in-depth interviews’. We have clarified this in our manuscript. Please see page 6.

Results:

Comment 3.23: -It is unclear how the breakdown of characteristics of the participants relate to PHC. The designation of type of provider is unclear as is the type of health facility. More details are needed.

Response 3.23: We have made revisions in table 1 and in the narration to bring to clarity. Furthermore, on page 5 & 6 background information related to primary health facilities are also added along with the role of different types of primary healthcare workers.

Comment 3.24: Data analysis: Indicates that content analysis is used for data analysis, although the presentation of findings don't necessarily appear to reflect content analysis.

Response 3.24: As advised, we have elaborated our data analysis section to bring more clarity – specifically regarding content analysis. Please see page 7 of the manuscript. We have also cited below article that guided our analytical approach.

Hsieh HF, Shannon SE. Three Approaches to Qualitative Content Analysis. Qual Health Res 2005; 15:1277-1288.<https://doi.org/10.1177/1049732305276687>

VERSION 2 – REVIEW

REVIEWER	Rachelle Ashcroft University of Toronto
REVIEW RETURNED	11-Nov-2022
GENERAL COMMENTS	It would be helpful to add more clarity on the Lady Health Workers in terms of directly stating that it is the name of a program, and by adding a reference that pertains specifically to the lady health workers such as this one: https://archpublichealth.biomedcentral.com/articles/10.1186/s13690-021-00541-3

VERSION 2 – AUTHOR RESPONSE

Response 2.1: Thank you for your advice. We have added more details specifically about the role of lady health workers in the manuscript. Please see lines 218 – 222 on page 6. Moreover, the programme has been introduced in the 'country context' paragraph on page 5 line number 193 – 195. Furthermore, the suggested article has also been cited.

VERSION 3 – REVIEW

REVIEWER	Rachelle Ashcroft University of Toronto
REVIEW RETURNED	17-Nov-2022
GENERAL COMMENTS	The authors have responded to the requested revisions.